# The Antithrombotic Function of Sphingosine-1-Phosphate on Human Adipose-Stem-Cell-Recellularized Tissue Engineered Vascular Graft In Vitro

**DOI:** 10.3390/ijms20205218

**Published:** 2019-10-21

**Authors:** Chih-Hsun Lin, Jen-Her Lu, Kai Hsia, Hsinyu Lee, Chao-Ling Yao, Hsu Ma

**Affiliations:** 1Division of Plastic and Reconstructive Surgery, Department of Surgery, Taipei Veterans General Hospital, Taipei 11217, Taiwan; chlin12@vghtpe.gov.tw (C.-H.L.); sma@vghtpe.gov.tw (H.M.); 2Department of Surgery, School of Medicine, National Yang-Ming University, Taipei 11221, Taiwan; 3Department of Pediatrics, Taipei Veterans General Hospital, Taipei 11217, Taiwan; hkay1008@gmail.com; 4Department of Surgery, medicine & Pediatrics, School of Medicine, National Defense Medical Center, Taipei 11490, Taiwan; 5Department of Pediatrics, School of Medicine, National Yang-Ming University, Taipei 11221, Taiwan; 6Department of Life Science, National Taiwan University, Taipei 10617, Taiwan; hsinyu@ntu.edu.tw; 7Department of Chemical Engineering and Materials Science, Graduate School of Biotechnology and Bioengineering, Yuan Ze University, Chung-Li, Taoyuan City 32003, Taiwan; s897601@oz.nthu.edu.tw

**Keywords:** sphingosine-1-phosphate, endothelial cell, ASC, platelet adhesion, syndecan-1

## Abstract

Adipose stem cells (ASCs) show potential in the recellularization of tissue engineerined vascular grafts (TEVGs). However, whether sphingosine-1-phosphate (S1P) could further enhance the adhesion, proliferation, and antithrombosis of ASCs on decellularized vascular scaffolds is unknown. This study investigated the effect of S1P on the recellularization of TEVGs with ASCs. Human ASCs were derived from lipoaspirate. Scaffolds were derived from human umbilical arteries (HUAs) with treatment of 0.1% sodium dodecyl sulfate (SDS) for 48 h (decellularized HUAs; DHUAs). The adhesion, proliferation, and antithrombotic functions (kinetic clotting time and platelet adhesion) of ASCs on DHUAs with S1P or without S1P were evaluated. The histology and DNA examination revealed a preserved structure and the elimination of the nuclear component more than 95% in HUAs after decellularizaiton. Human ASCs (hASCs) showed CD29(+), CD73(+), CD90(+), CD105(+), CD31(–), CD34(–), CD44(–), HLA-DR(–), and CD146(–) while S1P-treated ASCs showed marker shifting to CD31(+). In contrast to human umbilical vein endothelial cells (HUVECs), S1P didn’t significantly increase proliferation of ASCs on DHUAs. However, the kinetic clotting test revealed prolonged blood clotting in S1P-treated ASC-recellularized DHUAs. S1P also decreased platelet adhesion on ASC-recellularized DHUAs. In addition, S1P treatment increased the syndecan-1 expression of ASCs. TEVG reconstituted with S1P and ASC-recellularized DHUAs showed an antithrombotic effect in vitro. The preliminary results showed that ASCs could adhere to DHUAs and S1P could increase the antithrombotic effect on ASC-recellularized DHUAs. The antithrombotic effect is related to ASCs exhibiting an endothelial-cell-like function and preventing of syndecan-1 shedding. A future animal study is warranted to prove this novel method.

## 1. Introduction

Adipose stem cells (ASCs) show great potential in regenerative medicine, as they have the ability to differentiate into different lineages. ASCs may also exert a paracrine effect to modulate the host microenvironment [1]. Recently, ASCs have been considered in the field of tissue engineering technology for tissue or organ fabrication in vitro. The advantages of ASCs in tissue engineering are mainly related to the ease of harvesting a large amount of fat for cell cultivation, the lack of ethical concerns, and their pluripotency [2]. To date, ASCs have been used in tissue engineering research of bone, cartilage, adipose tissue, tendon, skin, and so forth [3,4,5,6]. They have also been proposed for use in vascular tissue engineering [7].

Endothelial cell (EC) recellularization is known specifically for the patency of tissue- engineered vascular grafts (TEVGs) in vivo [8,9]. In addition to ECs, ASCs have recently shown potential in recellularization of TEVGs [7,10]. It is known that ASCs could be induced to ECs or smooth muscle cells (SMCs) under specific conditions [11,12,13,14], but whether ASCs plays a role in replacing ECs in the recellularization of TEVGs still requires further study. The current ASC application strategy in vascular tissue engineering mainly encompasses two categories. One is the direct differentiation of ASCs into ECs or SMCs and then seeds these differentiated cells onto scaffolds for cultivation in vitro and then implanting them in vivo [15,16]. Another way is to seed ASCs or a stromal vascular fraction onto scaffolds under static or dynamic culture and then implant in vivo [17,18]. The first method supplies growth factors, such as vascular endothelial growth factor (VEGF), to stimulate EC signaling and expression in ASCs [15,16]. The main shortcoming of this method is that it is time-consuming and creates a risk of tumor transformation in the growth factor. In the second category, a dynamic culture, such as a bioreactor, could simulate in vivo the perfusion status to trigger mechanical receptor for shear stress, followed by a signal cascade for EC differentiation of ASCs [13,19]. The disadvantages of this method are that it is time consuming and expensive. 

Regarding clinical translation, an efficient recellularization process for TEVGs is important for achieving both the patient’s requirements and cost-effectiveness. Thus, the development of a rapid and effective recellularization process that simultaneously stimulates EC differentiation of ASCs and exhibits antithrombotic functions may be regarded as an ideal method for clinical translation of TEVGs.

Sphingosine-1-phosphate (S1P), a low-molecular-weight phospholipid, regulates diverse biological activities of EC and maintains EC barrier function that is correlated with vessel development [20,21,22,23,24]. In our previous work, we found that using S1P to enhance EC attachment in a tissue-engineered vascular scaffold and the S1P-augmented endothelium significantly enhanced antithrombotic surface properties due to reduced syndecan-1 (SDC-1) shedding [25]. Furthermore, we demonstrated that S1P enhances the patency of EC-recellularized vascular scaffolds in a rat aorta implantation model [26]. The main point of S1P on EC that could prevent platelet adhesion because it promotes SDC-1 expression and prevents SDC-1 shedding. The SDC-1 exhibition on cells was regarded as an indicator to predict the maturity and antithrombosis function of EC [25]. Recently, it has shown that S1P could promote the differentiation of ASCs toward endothelial-like cells, which could express endotheial nitric oxide synthase (eNOs) [11]. But whether S1P has a similar antithrombotic effect on ASCs in reconstitution of TEVGs is unknown.

Thus, in this study, we aimed to investigate if S1P could benefit ASC recellularization and express SDC-1, thereby decreasing blood clot formation or platelet aggregation on an ASC-based TEVG in vitro. Here, the ASCs were derived from human lipoaspirate. Human umbilical arteries were decellularized by 0.1% sodium dodecyl sulfate (SDS). ASCs with or withot S1P addition were seeded on decellularized human umbilical arteries (DHUAs). The potential effects of S1P on ASCs for EC differentiation and antithrombotic function were evaluated.

## 2. Materials and Methods

### 2.1. Ethics Assurance

Human umbilical cords and cord blood were harvested at Department of Obstetrics, Taipei Veterans General Hospital (Taiwan), with informed consent signed by the donors. The entire procedure was performed in accordance with governmental regulations (Guidelines for Collection and Use of Human Specimens for Research, Department of Health, Taiwan) and after approval from the Institutional Review Board (approval number 2013-08-020BC, Taipei Veterans General Hospital, Taiwan). Human ASCs were collected and established after approval from the Institutional Review Board of Taoyuan General Hospital, Ministry of Health and Welfare, Taiwan (approval number TYGH106078, 20, April, 2018, Taoyuan General Hospital).

### 2.2. Preparation of the DHUA (Decellularized Human Umbilical Arteries)

HUAs of approximately 5 cm in length were isolated by removing the veins and Wharton’s jelly. The decellularized process was done as previously described [25]. Briefly, it started with a 2-day incubation in 0.1% SDS (Sigma-Aldrich, St. Louis, MO, USA). Then, the grafts were washed with phosphate-buffered saline (PBS; pH 7.4; Gibco, Carlsbad, CA, USA) for another two days. The remaining DNA was deleted by medium 199 (Gibco) containing 20% fetal bovine serum (FBS; Gibco, South America) in the following two days. The vessels were washed with PBS subsequently. All the decellularization steps were performed at 37 °C on a shaker with high-speed agitation under sterile conditions. The cellularity and histomorphology of decellularized vessels were examined by hematoxylin and eosin (HE; Sigma-Aldrich), Dapi staining, Masson’s Trichrome and Elastin Van Gieson’s (EVG) staining. DNA quantification was done using Quant-iT™ PicoGreen^®^ dsDNA reagent (Invitrogen, Carlsbad, CA, USA) to confirm that more than 95% of the cells were removed and the complete scaffold structure remained intact [27]. DNA quantification was performed according to the manufacturer’s instructions. Briefly, native and decellularized umbilical arterial segments were dried, weighed, and digested in papain buffer (papain 125 ug/mL Sigma) at 60 °C overnight. Papain (papaya proteinase I) is a cysteine protease enzyme that breaks peptide bonds and can be used to digest tissue, and the nuclear acid can be released after digestion. The papain-digested sample was diluted with TE buffer (10 mM Tris-HCl and 1 mM EDTA, pH 7.5; Invitrogen) and incubated with PicoGreen reagent. Using a fluorometer, the fluorescence was measured at an excitation wavelength of 485 nm and an emission wavelength of 530 nm. Bacteriophage λ DNA (Invitrogen) was used as a standard [28]. The quantification of DNA was done in triplicate.

### 2.3. Isolation, Culture, and Characterization of hASC and Human Umbilical Vein Endothelial Cells (HUVEC)

#### 2.3.1. The hASC (Human Adipose Stem Cells)

The adipose tissue was harvested by means of a standard liposuction procedure, and then it underwent a centrifugation of 300× *g* for 10 min, in order to remove the oil and serous fractions. Then, hASC was released by type IV collagenase (Sigma-Aldrich) digestion from the surrounding connective tissue scaffold for 30 min at 37 °C. Finally, a double series of washing with PBS and centrifugation was conducted to obtain the hASC pellet. The hASC was then cultured on a 10 cm plate in DMEM medium containing 10% FBS and subcultured every two to three days, and passages five to seven were used in the experiments.

HUVECs and hASCs were identified by surface markers, including human CD29, CD90, CD34 (Abcam, Cambridge, MA, USA), CD31, CD44, CD73, CD105, CD146, HLA-DR (Becton–Dickinson, San Jose, CA, USA), using a BD FACSCanto Flow Cytometer (Becton–Dickinson, San Jose, CA, USA). A replicate unstained sample was used as a negative control. Data were analyzed with the BD FACSDiva Version 6.1.3 and the FlowJo 10.1 software (BD Biosciences, Ashland, OR, USA). The differentiation potential of ASCs to adipogenic, osteogenic, and chondrogenic linage was examined using a differentiation–induction protocol and differentiation assay described previously [29,30]. 

#### 2.3.2. HUVEC

HUVECs were isolated from fresh umbilical cords by treatment with 0.1% type I collagenase (Sigma-Aldrich) in cord buffer (136.9 mM NaCl, 4 mM KCl, 10 mM HEPES, and 11.1 mM glucose pH7.65) in a 37 °C incubator for 20 min. After incubation, HUVECs were collected by centrifugation 1500× *g* for 5 min and cultured on 10 cm plates in Endothelial Cell Growth Medium (EGM) (Cell application, San Diego, CA, USA), containing 10% fetal bovine serum, penicillin (100 U/mL), and streptomycin (100 mg/mL). Cells underwent one passage weekly and were subcultured after trypsinization. Passages three to five were used in the experiments. HUVECs were labeled with Dil-Ac-LDL (acetylated low-density lipoprotein labeled with l,l′-dioctadecyl-l-3,3,3′,3′,-tetramethylindocarbocyanine perchlorate) and stained with CD31 (Abcam) to confirm its characterization.

### 2.4. Adhesion and Proliferation of hASCs on DHUA with S1P

DHUA segments were cut into open patches and put into 12-well plates with the luminal surface facing up. The hASCs and HUVECs were stained with Cell TrackerTM CM-Dil (Invitrogen) before seeding. According to the manusfacture’s instructions, cells were incubated in the working solution with CM-DIL for 5 min at 37 °C, and then for an additional 15 min at 4 °C. Finally, cells were washed with PBS and changed to fresh medium. DHUA patches were then seeded with 4 × 10^5^ cells/mL of CM-Dil labeled hASCs or HUVECs, and 1 mL of EGM was added to each well. EGM was premixed with 1 uM S1P or 0.1% fatty-acid-free boine serum albumin (FAF-BSA) before being added to the cells. Cells were then incubated for 48 h at 37 °C and 5% CO_2_. After static cell seeding, the cellularity was visualized by fluorescence microscopy at 517 nm excitation, and the fluorescent cells were analyzed using the MetaMorph program (Molecular Devices, Sunnyvale, CA, USA). Some cell-seeded DHUAs were also processed as frozen sections to confirm the attachment of the cells. Each experiment was performed in triplicate.

### 2.5. Antithrombotic Assay for TEVGs (Tissue Engineerined Vascular Grafts)

#### 2.5.1. Reconstitution of TEVGs In Vitro

TEVGs were constructed by DHUA, HUVEC/hASC and S1P by the static seeding method. The cultured HUVECs at passages three to five or hASC at passages five to seven were harvested and diluted with 1 mL of EGM culture medium and 1 μM S1P. The 0.1% FAF-BSA served as solvent control. The cell suspension (5.0 × 10^7^ cells/mL, 0.2 mL) was injected into the lumen of DHUA by a 1 mL syringe to facilitate dilatation of the lumen (length, 1 cm; inner diameter, ~2–3 mm), and the DHUA was then placed in a 6-well plate with the two ends clamped by microvessel clamps. The grafts was immersed in the culture medium for 6 h at 37 °C, while being rotated 360° around its longitudinal axis. Another cell suspension was then added in the same manner, and the graft was rotated 90° around its longitudinal axis. After repeating this procedure four times (total seeded cells, 4.0 × 10^7^ cells), the graft was incubated for another 24 h to ensure complete cell attachment. After two days of static incubation in EGM, the cell suspension in the cultured tube was renewed with fresh medium and incubated for another 24 h to ensure complete cell attachment. The new generated grafts then underwent antithrombotic functional assay, including kinetic clotting time and platelet adhesion.

#### 2.5.2. Coagulation and Kinetic Clotting Tests

Venous blood samples were drawn from the antecubital vein of healthy volunteers and collected in a 15 mL tube containing 3.8% sodium citrate (Sigma-Aldrich) as an anticoagulant at a 1:9 ratio of sodium citrate/blood. The blood was injected into the TEVGs containing hASC or HUVEC in the presence or absence of 1 µM S1P. Non-cell-seeded DHUAs served as controls. To initiate the blood coagulation cascade, 0.25 M of calcium chloride solution was added to the citrated blood samples. The two ends of the 1 cm vessels were then sealed, and, after a predetermined time, one end of each artery was cut, and the blood sample was transferred into a 15 mL tube containing 5 mL of distilled water. The lumen of the vessels were also washed by the water in order to lyse the erythrocytes adherence on the wall. Red blood cells were broken up by a hypotonic solution and released hemoglobin. The red blood cells that were not trapped in a thrombus were hemolyzed, whereas free hemoglobin was dissolved in water. The concentration of the free hemoglobin dissolved in water was colorimetrically measured at 540 nm wavelength using a plate reader. The change in the optical density of the solution versus time was plotted. Clotting times were estimated for all test materials, including non-seeded DHUA, cell-seeded DHUA, and cell-seeded DHUA with S1P treatment, as previously described [28,31]. Each experiment was performed in triplicate. In addition, the tissue samples at each time-point were also sent for H&E staining to check thrombus formation directly.

#### 2.5.3. Platelet Adhesion Test

Platelet-rich plasma (PRP) was obtained from healthy donors by centrifugation at 200× *g* for 20 min, at room temperature, without braking. Approximately two-thirds of the PRP was transferred into a new plastic tube and centrifuged at 100× *g* for 20 min, at room temperature, without braking to pellet contaminating red and white blood cells. The supernatant was again transfered into a new plastic tube and centrifuged at 800× *g* for 20 min, at room temperature, without braking to pellet platelets. The CD62 positive platelet was about 40% in the PRP separated by a similar method [32]. The pellet was suspended in D-PBS at a concentration of 1.0 × 10^9^ mL^−1^. Then, platelet adhesion on native artery, DHUA, and TEVGs, on which were seeded by CM-DIL labeled cells, approximately 1 cm in length, was evaluated. Six samples for each group were immersed in PRP, incubated at 37 °C for 1 h, and subsequently rinsed with a 0.9% NaCl solution (Sigma-Aldrich) to remove weakly adherent platelets. Next, the adherent platelets were fixed with 2.5% glutaraldehyde solution (Sigma-Aldrich) at room temperature for 16 h. The samples were washed three times for 10 min each with D-PBS and then treated with 1% osmium tetroxide (Sigma-Aldrich) for 1 h. After treatment, the samples were rinsed with D-PBS three times before embedding for SEM examination. The specimens were coated with a 10–20 nm thick gold layer after critical-point drying and examined using SEM. Ten fields at 2000× magnification were chosen at random to obtain high-confidence statistics to quantify adherent platelets. In the meantime, after PRP adhesion, part of the grafts were taked to fixed in 4% paraformaldyhide for 30 min and processed frozen section to further confirm the cells on the substrates. Fluorescent images were obtained with a Zeiss Optimises Axio Imager A1 fluorescence microscope (ZEISS, Jena, Germany). 

### 2.6. The Effect of S1P on the Expression of Syndecan-1 and CD 31 of hASCs 

The hASCs were cultured at a density of 500 cells/cm^2^ in EGM, with 1 uM S1P or 0.1% FAF-BSA (Sigma-Aldrich) as solvent control at 5%CO_2_ and 37 °C for 14 days. The same treatment was done on HUVECs as positive control. For EC differentiation, the characteristics of hASCs were identified according to 2.3.1. For expression of CD31 or syndecan-1, The cells were then fixed in 1% paraformaldehyde/PBS and blocked in PBS with 1% BSA for 30 min and then incubated with primary antibodies syndecan-1 (Abcam) and CD 31 (Abcam) at 4 °C overnight. After being washed in PBS, the cells were incubated with the second antibody goat Alexa Fluor 488 anti-mouse IgG (Thermo Fisher Scientific, Waltham, MA, USA) at room temperature for 1 h. The nuclei were counterstained with DAPI (Thermo Fisher Scientific). Fluorescent images were obtained with a Zeiss Optimises Axio Imager A1 fluorescence microscope. The experiement was performed in triplicate. While the images were captured, a primary-free and only secondary antibody control stained sample of each antibody was applied to set the exposure time in order to rule out a nonspecific binding of the fluorescent secondary to our syndecan-1 or CD 31 stained samples.

### 2.7. Statistical Analysis

All experiment data were shown as the mean ± SD. The Mann–Whitney U test was applied for statistical analysis between experimental groups, where *p* < 0.05 was considered significant. The Kruskal–Wallis test with the post hoc Mann–Whitney U test was used to compare data of more than two groups, where *p* < 0.05 was considered significant. Statistical analysis was performed using IBM SPSS Statistics 19 (version 19; SPSS, Chicago, Illinois). 

## 3. Results

### 3.1. Characterization of DHUA

The results showed DHUA preserved tubular structure grossly but turned whitish in appearance (Figure 1A: native, Figure 1B: DHUA). H&E staining revealed a decrease of most nuclei in DHUA (Figure 1C: native, Figure 1G: DHUA). Masson’s Trichrome and Elastin Van Gieson’s staining indicated that most of the fibrous structure was preserved in layers (Figure 1D,E: native, Figure 1H,I: DHUA). Dapi staining confirmed the depletion of nuclear component after decellularization (Figure 1F: native, Figure 1J: DHUA); DNA quantification also showed that more than 95% DNA was removed from the HUA (Figure 1K, *p* < 0.01). 

### 3.2. Characterics of hASC and HUVEC

In this study, hASCs or HUVECs were seed on DHUAs to compare their effects in the present of S1P on the constructuin of TEVGs. Before cell seeding, the characterization of hASCs and HUVECs was identified, including their cell morphology, surface marker expression, and Dil-Ac-LDL uptake test. The hASCs were fusiform-spindle in morphology (Figure 2A) and CD29(+), CD73(+), CD90(+), CD105(+), CD31(−), CD34(−), CD44(−), CD146(−), and HLA-DR(−) in surface marker analysis (Figure 2B). HUVECs, on the other hand, were cobblesteon in morphology (Figure 2C), Dil-Ac-LDL (+) (Figure 2D) CD31(+), and CD34(−) (Figure 2E). The results of adipogenic, osteogenic, and chondrogenic differentiation of hASCs are shown in Appendix A.

### 3.3. The Effect of S1P on Adhesion and Proliferation of hASCs on DHUA

As displayed in Figure 3, the results showed that CM-Dil-stained cells could adhere and proliferate on DHUA. HUVECs could adhere on DHUA (left panel, first row). The number of adhesive HUVECs on DHUA increased after two days of S1P treatment (left panel, second row). Fluroscence intensity showed a significant increase in proliferation (cell count) of HUVECs on DHUA (right panel, upper, *p* < 0.01). hASCs could also adhere on DHUA (left panel, third row), but the number of cell adhesions did not increase significantly after two days of S1P treatment (left panel, fourth row). Fluroscence intensity did not reveal a significant increase of proliferation (cell count) of hASCs on DHUA (Right panel, lower, *p* > 0.05). These results demonstrated that S1P can enhance HUVEC attachment and proliferation to the DHUV scaffolds but did not effect that of hASC. When the treatment was continuous to day 14, both HUVEC and hASC reached about 80% coverage to the DHUA, and there was no significantly difference between the groups in the presence of S1P and not (please see Appendix A). 

### 3.4. The Effect of S1P on Coagulation and Kinetic Clotting Time of hASC-Recellularized DHUA

The kinetic clotting test showed two days of S1P-treated hASC-recellularized DHUA (Figure 4A) could significantly decrease blood clotting at 25, 30, 45, and 50 min (*p* < 0.05, *p* < 0.01, *p* < 0.01, and *p* < 0.01, respectively). In order to further assess the anti-thrombotic effect of TEVGs directly, H&E stain was applied to evaluate thrombus formation in these samples. The results exhibited quick blood coagulation with thrombus formation in the lumen of DHUA (Figure 4B–E). The degree of thrombus formation was significantly lower on S1P-treated hASC-recellularized DHUA (Figure 4J–M) in comparison to those untreated hASC-recellularized DHUA, at 25, 30, 45, and 50 min (Figure 4F–I).

### 3.5. The Effect of S1P on the Platelet Adhesion of hASC-Recellularized DHUA

As shown in Figure 5, when comparison of HUA and DHUA, the SEM images showed more platelet adhesion on DHUA (38.1 ± 12.3/field, Figure 5B) than HUA (3.1 ± 2.0/field, Figure 5A). The platelet count revealed significantly more platelet particles on DHUA (Figure 5G, *p* < 0.01). When comparing HUVEC-recellularized TEVGs with or without S1P, the SEM images showed more platelet adhesion on HUVEC-recellularized TEVGs without S1P (5.0 ± 2.1/field, Figure 5D and 11.2 ± 3.1/field, Figure 5C). The platelet count revealed significantly fewer platelet particles on HUVEC-recellularized TEVGs with S1P (Figure 5H, *p* < 0.05). When comparing ASC-recellularized TEVGs with or without S1P, the SEM images showed more platelet adhesion on ASC-recellularized TEVGs without S1P (13.2 ± 5.2/field, Figure 5F and 17.0 ± 5.3/field, Figure 5E). The platelet count revealed significantly fewer platelet particles on ASC-recellularized TEVGs with S1P (Figure 5I, *p* < 0.05). The images in Figure 5J–Y presented that the cells, including HUVEC (Figure 5J–M for FAF-BSA treated HUVEC, Figure 5N–Q for S1P treated HUVEC) and hASC (Figure 5R–U for FAF-BSA treated hASC, Figure 5V–Y for S1P treated hASC) were ensured to be cultivated on DHUA.

### 3.6. The Effect of S1P on EC Differentiation and SDC-1 Expression of hASCs In Vitro

To investigate endothelial differentiation potential of hASCs under S1P treatment, we analyzed endothelial differentiated cells by flow cytometry analysis, morphology, and IHC staining (CD31). The results of flow cytometry showed that hASCs turned on CD31 expression and turned off CD29, CD73 expression after five days of culture in S1P-added medium (Figure 6A). The expression of CD90 and CD105 on hASCs were less than that of cells before treatment (Figure 6C). The hASCs had profiles similar to endothelial-lineage-associated surface markers. The morphology of hASCs became rounder under S1P-added medium, although the cells still did not show typical ECs (Figure 6B). The immunohistochemical stains revealed positive CD31 staining in hASCs (Figure 6D–F, D: CD31, E: DAPI, F: merge). The results indicated that S1P induced hASCs differentiate toward endothelial lineage.

Regarding SDC-1 expression, S1P increased SDC–1 expression in HUVECs (Figure 7A–C: cultured HUVECs without S1P and Figure 7D–F: cultured HUVECs with S1P). Fluorescent intensity revealed significant increase of SDC-1 of HUVECs under S1P treatment (Figure 7M: the fluorescence intensity of SDC-1 expression on cultured HUVECs without and with S1P, *p* < 0.01). Similary, S1P increased SDC-1 expression in hASCs (Figure 7G–I: cultured hASCs without S1P and Figure 7J–L: cultured hASCs with S1P). Fluorescent intensity revealed significant increase of SDC-1 of hASCs under S1P treatment (Figure 7N, the fluorescence intensity of SDC-1 expression on cultured hASCs without and with S1P, *p* < 0.01).

## 4. Discussion

The development of an antithrombotic inner surface is criticial for TEVGs. In the past, complete endothelialization was regarded as the key factor for the patency of TEVGs [33]. However, there is no concensus on what degree of endothelialization is complete enough to avoid thrombus formation once under blood flow. Regardless, many factors exist for the recellularization of TEVGs, such as cell type, capability of adhesion and proliferation on TEVGs, cell number, culture environment, culture duration, and so forth [9]. In addition, there is an interaction between the inner surface of TEVGs, either synthetic or biological, and blood cells, even without complete coverage of ECs. Although initial thrombi formation at the inner surface of TEVG is expected, resolving the thrombus is possible if an antithrombosis microenvironment is established mostly at the inner surface of TEVGs [34,35].

Regarding the functionalization of the inner surface of TEVGs, many strategies have been proposed to modify or resurface the blood-material interface with biopeptides, growth factors, or exosomes, among others [36,37]. The concept focuses on recruiting endothelial progenitor cells or monocytes/macrophages in the blood flow to reside on the interface, followed by the endothelialization process, as a barrier to resist platelet aggregation, break coagulation cascade, and assist vasodilatation [38,39]. This early adapted phase in blood-material contact is crucical for the patency of blood flow and permits the proceeding degradation of scaffolds, neo-tissue formation, and tissue remodeling [40].

Due to concerns over cost effectiveness, both reducing the time of in vitro culture/expansion/differentiation and enhancing efficient thrombus resistance are important for developing TEVGs [35]. To improve the patency of TEVGs, efforts should focus on recellularization, preconditioning, and surface modification. However, the kind of cells-progenitor cells or mature cells for recellularization is not fully determinant [9]. At first, the approach of using stem cells in TEVGs required differentiation of the stem cells into ECs or SMCs, which were then seeded on scaffolds. However, this process is time-consuming, and the fate of seeded cells remained controversial. For example, Row et al. indicated that host cellular infiltration was dominant in TEVGs, while the donor cells (ECs/SMCs) were gradually replaced by host cells within three months of implantation [41]. Kurobe et al. also indicated that seeded cells do not differentiate and populate the neovessel; instead, seeded cells play a pivotal role in promoting cellular migration and ingrowth from circulating/adjacent host cells [40]. Hence, cell selection for TEVG recellularization could be broadened to adult stem cells, such as ASCs, due to their advantages in clinical translation. Recently, ASCs or stromal vascular fractions (SVFs) were also proposed for the recellularization of TEVGs and have shown potential for differentiation into ECs and SMCs [7,17]. Dimuzio et al. noted the possibility of using ASCs in TEVG creation and suggested the potential of these cells to become endothelial-like under shear force in vivo [7]. The application of ASCs or SVFs aims at reducing timing for cell differentiation and more closely meeting the requirements of clinical patients. Thus, our study goal was to apply ASCs in TEVGs and extend the field to investigate the effect of biolipids on recellularization and antithrombosis.

Presently, few studies are conducted on the incorporation of S1P with ASCs in TEVG development. However, some studies have already discussed the relationship between S1P and the bioactivity of ASCs [42,43,44]. S1P has pleiotropic bioactivity, which can regulate various cellular activities through five specific receptors (S1P1-5) [45]. It was already shown that S1P can stimulate ASC differentiation into ECs or SMCs. Arya et al. found that S1P could promote the expression of eNOS in ASCs and push ASC differentiation toward endothelial-like cells. We also found that S1P-treated ASCs would show CD31 expression, which was initially negative. The effect of EC differentiation on ASCs by S1P was mainly through the S1P1 receptor to upregulate PI3K [11], though some studies have shown the proliferative benefit of S1P on ASCs; for example, Shen et al. showed biolipid could promote proliferation of ASCs through the S1P1 receptor and Akt/ERK1/2 pathway [46]. Marycz et al. showed that S1P increases ASC proliferation by the formation of cellular aggregates [47]. However, on decellularized vascular scaffolds, we did not find that S1P significantly increased proliferation of ASCs. The reason could be related to EC differentiation of ASC under S1P treatment. As cells differentiated, their rate of proliferation usually decreases [48].

Moreover, Nincheri et al. revealed S1P could dose-dependently stimulate ASCs toward smooth muscle cell differentiation. However, S1P2 receptor plays a criticall role in this effect. In their study, the minimum concentration of S1P used to induce differentiation of ASC toward smooth muscle cell was 0.1 uM (after six days). Under 1 uM S1P treatment for six days, profound cytoskeletal reorganization was noted in ASCs [49]. However, culture medium used for smooth muscle differentiation was DMEM without serum containing BSA and not endothelial cell growth medium used in our study. Furthermore, the S1P1 receptor is also related to the protection of glycocalyx from shedding by metalloproteinases [50]. Our results also indicated that S1P could stimulate a significant increase in the expression of syndecan-1 on ASCs in a way that was similar to HUVECs. In addition, adipose stem cell, as one of the mesenchymal stem cells, can secrete angiogenic, chemotactic, and mobilizing factors [51]. Endothelialization of TEVGs also depends on recruit host cells onto the graft by paracrine signaling [52]. The immune modulatory effect of ASCs also cloud have advantages for biomedical implants to reduce the foreign-body reaction [53].

Considering the abovementioned findings and our preliminary in vitro results together, we propose that S1P could potentiate the antithrombotic effect of ASC-recellularized TEVGs. Furthermore, ASCs could play a role as temporal cellular coverage on the inner surface of TEVGs, instead of direct EC recellularization. Although the true cost-effectiveness of using ECs or ASCs for recellularization still needs to be evaluated, our results showed that two to five days of S1P treatment could possibly enhance ASC toward EC differentiation. The antithrombotic effect could be persistent within 14 days of S1P treatment. In addition, S1P is related to many EC function, such as proliferation, cellular barrier integrity, maintaining vascular permeability, and signal transmission [21,54,55,56,57,58]. Similar to HUVECs, SDC-1 shedding from ASCs could be a component of thrombus formation when using ASCs for TEVG recellularization. S1P may therefore be useful in the construction of antithrombotic, ASC-based TEVG by preventing SDC-1 shedding.

## 5. Conclusions

This study investigated the antithrombotic effect of S1P on ASC-recellularized TEVGs in vitro. The results showed that reconstituted TEVGs decreased blood clotting and resisted platelet adhesion. These advantages could be related to S1P stimulating ASCs to perform EC-like function and preventing SDC-1 shedding. A future animal study is warranted to prove this novel concept.

## Figures and Tables

**Figure 1 ijms-20-05218-f001:**
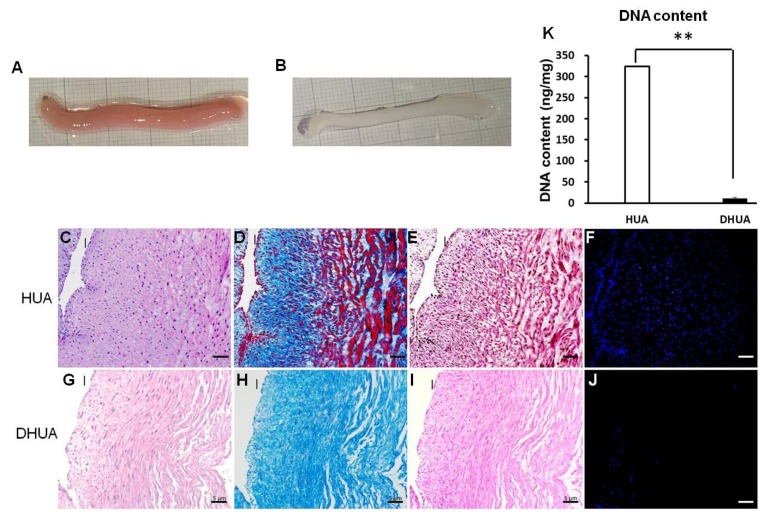
The characterization of DHUA. (**A**,**B**) Gross appearance of HUA and DHUA; (**C**–**F**) H&E, Masson’s, EVG, and DAPI staining of HUA; (**G**–**J**) H&E, Masson’s trichrome, EVG, and DAPI staining of DHUA (magnification 200×, scale bar = 5 μm); (**K**): DNA content of HUA and DHUA, which showed that more than 95% DNA was removed from the HUA. ** *p* < 0.01. I: intima.

**Figure 2 ijms-20-05218-f002:**
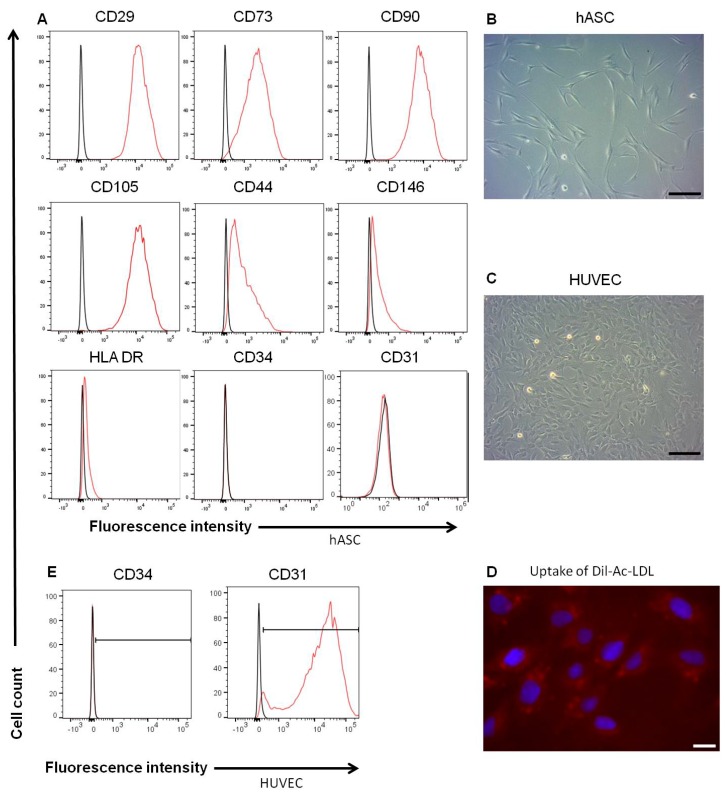
The Characteristics of hASCs and HUVECs. (**A**) The surface markers of hASC. (**B**) The morphology of hASC under bright field (magnification 100×, scale bar = 200 μm). (**C**) The morphology of HUVEC under bright field (magnification 100×, scale bar = 200 μm). (**D**) Dil-AC-LDL uptake by HUVEC (magnification 400×, scale bar = 2 μm). (**E**) The surface markers of HUVEC.

**Figure 3 ijms-20-05218-f003:**
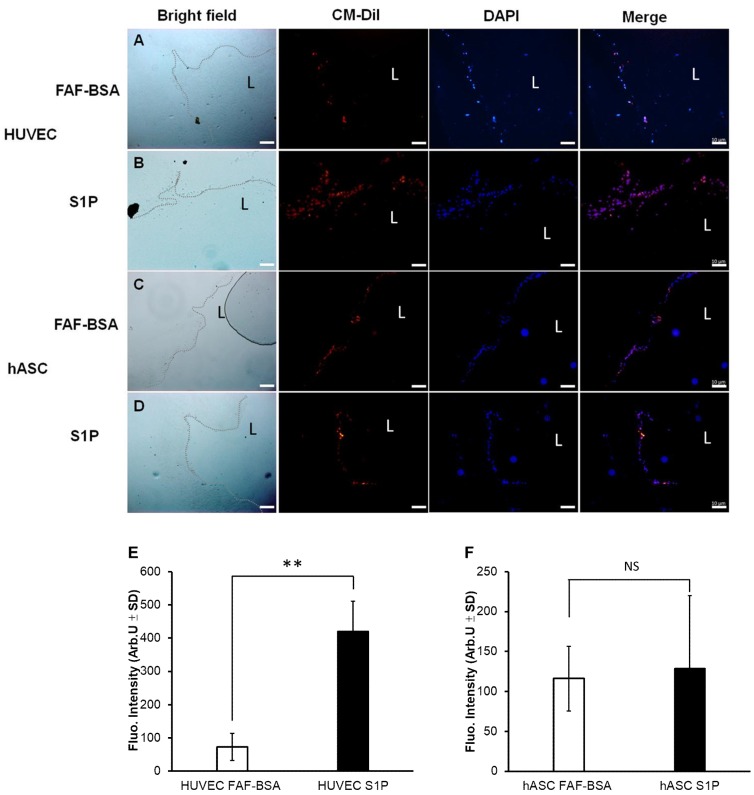
Adhesion and proliferation of hASC and HUVEC on DHUA with S1P. Cells prestained by CM-Dil seeded on DHUA under two days of S1P treatment. Upper panel, first column: bright field (dotted line indicates the foci of the cells). Second column: CM-Dil. Third column: DAPI. Fourth column: merge. First row (**A**): images of HUVECs seeded on DHUA without S1P; second row (**B**): images of HUVECs seeded on DHUA with S1P; third row (**C**): images of hASCs seeded on DHUA without S1P; fourth row (**D**): images of hASCs seeded on DHUA with S1P. Lower panel, left (**E**): fluorescence intensity of CM-Dil in HUVEC-seeded DHUA without and with S1P; right (**F**): fluorescence intensity of CM-Dil in hASC-seeded DHUA without and with S1P (all magnification 100×, Scale bar = 10 μm; ** *p* < 0.01, NS: not significant.). L: Lumen.

**Figure 4 ijms-20-05218-f004:**
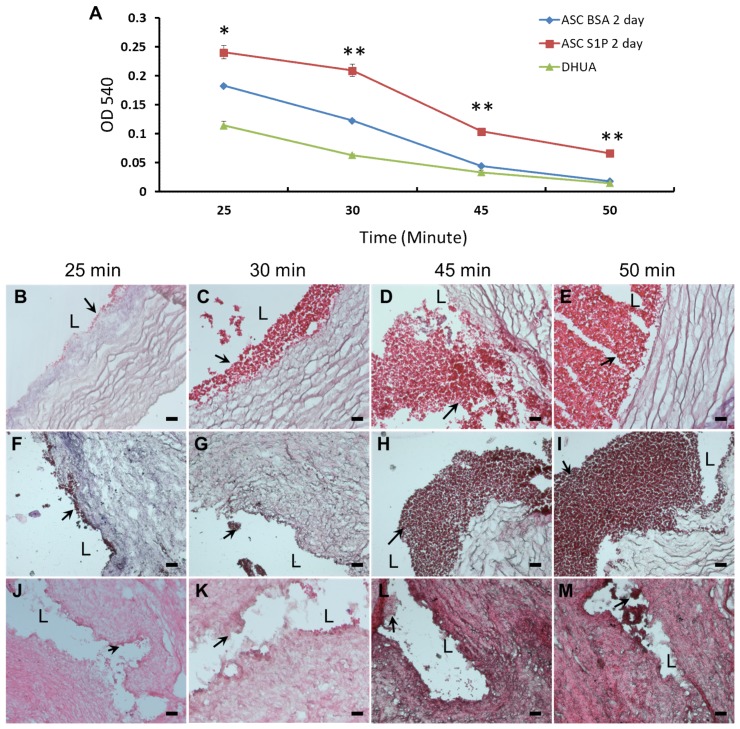
The effect of S1P on kinetic coagulation time of ASC-recellularized TEVGs in vitro. (**A**) The curve of kintic clotting time, the ASC-recellularized TEVGs with S1P treatment for 2 days showed significantly longer blood coagulation time than without S1P treatment or DHUA (* *p* < 0.05, ** *p* < 0.01). (**B**–**M**) The H&E staining of grafts after whole blood was injected at different time points. It showed the degree of thrombus formation. (**B**–**E**) DHUA; (**F**–**I**) hASC seeded on DHUA without S1P; (**J**–**M**) hASC seeded on DHUA with S1P. Black arrow points to thrombus. L: Lumen. Magnification 200×, scale bar = 5 μm.

**Figure 5 ijms-20-05218-f005:**
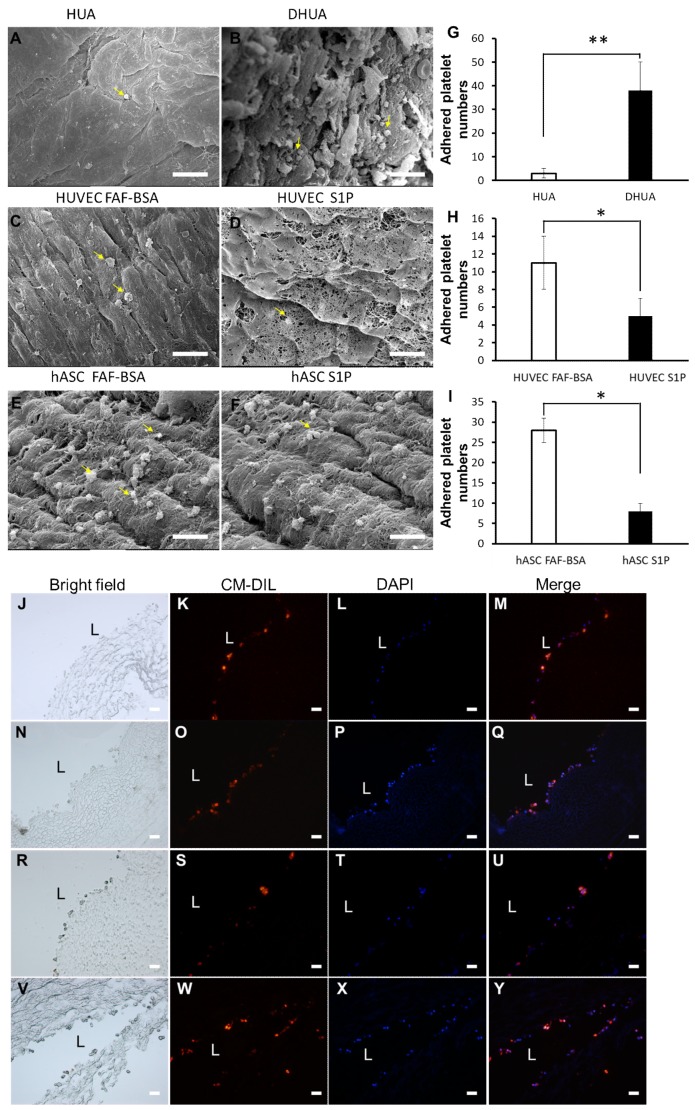
The effect of S1P on platelet adhesion of ASC-recellularized TEVGs in vitro. SEM images of platelet adhesion on HUA (**A**), DHUA (**B**), HUVEC-recellularized DHUA (**C**), HUVEC/S1P recellularized DHUA (**D**), ASC-recellularized DHUA (**E**), and ASC/S1P recellularized DHUA (**F**). The statistical results of platelet count on HUA, DHUA (**G**) and TEVGs (**H**,**I**). Yellow arrows point to platelets. (**G**) HUA showed significantly less platelet adhesion than DHUA. (**H**) HUVEC/S1P- recellularized TEVGs showed significantly less platelet adhesion than HUVEC recellularized TEVGs. (**I**) ASC/S1P-recellularized TEVGs showed significantly less platelet adhesion than ASC-recellularized TEVGs. (magnification 2000×, scale bar = 10 μm, * *p* < 0.05, ** *p* < 0.01). (**J**–**Y**) presented that the labeled cells were seeded on DHUV. (**J**–**M**) FAF-BSA treated HUVEC; (**N**–**Q**) S1P treated HUVEC; (**R**–**U**) FAF-BSA treated hASC; (**V**–**Y**) S1P treated hASC. (Magnification 200×, scale bar = 5 μm.).

**Figure 6 ijms-20-05218-f006:**
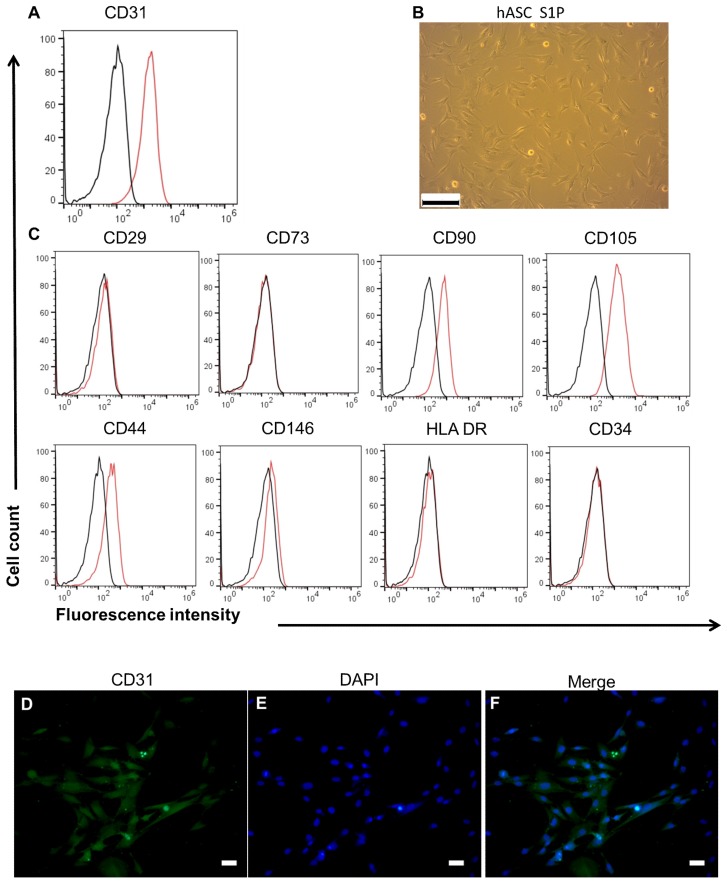
S1P induced change in surface markers and morphology of hASC. (**A**,**C**) After culture in S1P-added culture medium, hASCs showed positive CD31 and negative CD29, CD73. The expressin of CD90 and CD105 on hASC became less. It indicated that hASCs differentiated toward EC. (**B**) The morphology of hASCs under S1P-added culture medium (magnification 100×, scale bar = 200 μm). (**D**–**F**) The immunofluorescence stain of hASC by anti-CD31. (**D**) CD31; (**E**) DAPI; and (**F**) merge. (magnification 200×, scale bar = 5 μm).

**Figure 7 ijms-20-05218-f007:**
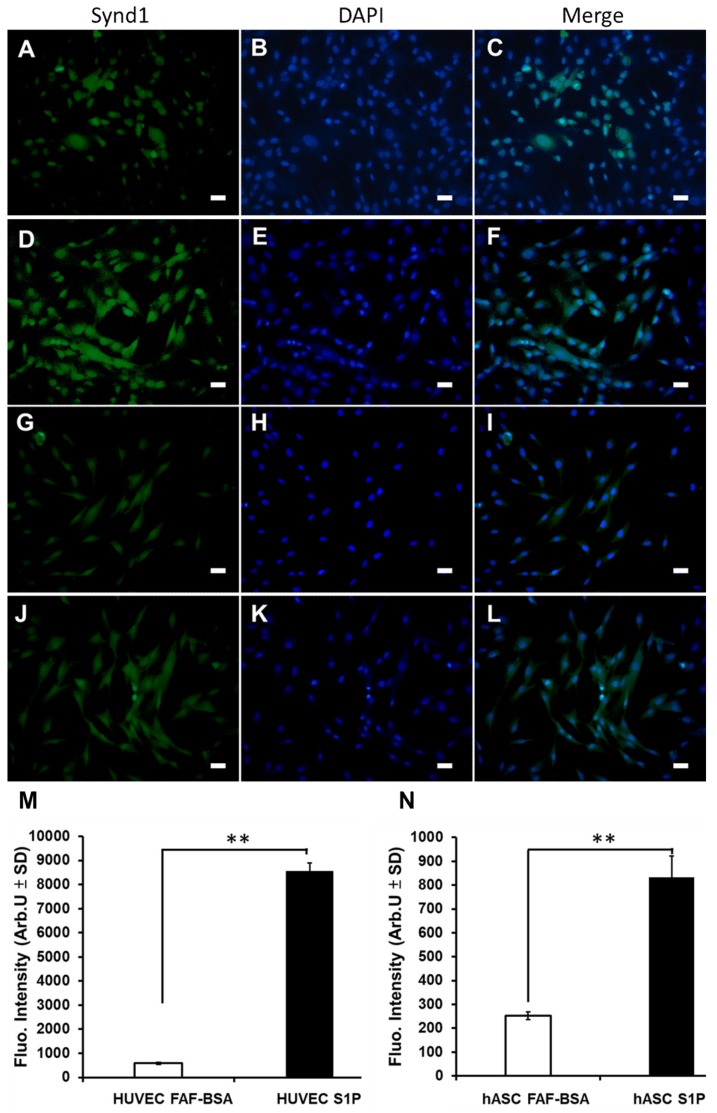
The effect of S1P on Sydecan-1expression in hASC in vitro. First column: SDC-1. Second column: DAPI. Third column: merge. (**A**–**C**) Cultured HUVECs without S1P; (**D**–**F**) cultured HUVECs with S1P; (**G**–**I**) cultured hASCs without S1P; (**J**–**L**) cultured hASCs with S1P; (**M**) upper—the fluorescence intensity of SDC-1 expression on cultured HUVECs without and with S1P; (**N**) the fluorescence intensity of SDC-1 expression on cultured hASCs without and with S1P. (All magnification 200×, scale bar = 5 μm; ** *p* < 0.01).

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
