# Peer review of "The Antithrombotic Function of Sphingosine-1-Phosphate on Human Adipose-Stem-Cell-Recellularized Tissue Engineered Vascular Graft In Vitro"

_ijms, 2019, doi:10.3390/ijms20205218_

Round 1

Reviewer 1 Report

In this paper the authors attempt to examine the impact of sphingosine-1-phosphate on human adipose stem cell recellularisation of decellularised human umbilical arteries.   The data doesn't appear to convincingly  show that hASC can effectively recellularise the DHUA, that the cell-free DHUA is triggering thrombus formation (and thus that hASC block this effect), hASC differentiation to endothelial cells, or that hASC have changes in syndecan expression in response to S1P, or that this is responsible for the asserted anti-thrombotic effect of hASC.  This may be due to the difficulties in my reading of the paper due to the poor English and incomplete figure legends. However  I don't think the conclusions are supported by the date, and would therefore not recommend publication of this articles in the International Journal of molecular sciences at this time.

Major comments. 

1. The quality of English is very poor throughout which makes assessing this paper effectively very difficult.  Their are spelling and grammatical errors in most lines.  This work needs significant rewriting before it is ready for publication  (e.g. in vitro is spelt incorrectly in the title, magnificence is used instead of magnification, constructuin instead of construction).  Figure legends are not always detailed enough to understand what is occuring in the experiments shown.

2. Figure 3 - The demonstration of hASC attachment to segments of the decellularised artery is very difficult to interpret.  The work appears to suggest that only a small percentage of the construct has adherent cells attached to it.  It would seem likely that the coverage of the DHUA is pretty minimal at day 2, and there is no similar data shown for day 14.  Without significant coverage of the DHUA it is unlikely that this construct would ever be viable as an implantable model.  Do the authors have other pictures to demonstrate the % of surface coverage achieved? Can they compare to HUVECs to see if these are better cells for tissue engineering as hypothesised in the introduction?  

3. Figure 4 - the clotting assay is indirect.  It relies on haemoglobin washed from the sample being solely due to thrombus formation.  However the differences may also be due to less adherence of erythrocytes directly to the surface of the DHUA.  This makes it difficult to assess an anti-thrombotic effect using this experiment.  A better, simpler demonstration would be to show the degree of thrombus formation seen either using fluorescent labels of platelets, fibrin and erythocytes, or by histological section on the construct, rather than using an indirect demonstration.

4. Can you observe hASCs in your SEM pictures in figure 5?  

5. Figure 5 -  Although there appears to be a single erythrocyte adherent to the surface in the SEM, there doesn't seem to be an obvious thrombus shown on the cell-free DHUA. How thrombogenic is the DHUA normally? Can you see large scale clotting with a light or fluorescent microscope?  If it doesn't trigger observable blood clotting,it would question how significant the anti-thrombotic effect of the hASC is?  Given the minimal surface coverage of hASC seen in Figure 3, I am unconvinced this is a significant biological effect of S1P in this biological system.

6. Figure 6C - appears to show CD90 and CD105 are higher in S1P-treated cells, even though the text says they are "turned off".  It is hard to judge this properly though as there is no text in the figure legend to explain what is being shown.  How endothelial-like do the authors consider these s1P-treated hASC cells?  Figure 6B doesn't show cells bearing the cobblestone morphology I would expect of endothelial cells, and appears to be solely on the basis of an increase in CD31 expression. 

7. Figure 7.  HUVECs have a relatively low level of basal syndecan expression across all the cells overlapping with the DAPI stain (7A), which is increased by S1P (7b).  In contrast, hASC appear to have limited basal expression of syndecan1 and this appears in discrete spots not overlapping with DAPI staining (7C). Although enhanced in 7D, it still doesn't correlate with all cells stained.  It would appear that this might be either non-specific binding of the secondary or there are heterogenous cell populations with differential expression of syndecan-1.  Given this, it needs exploring is difficult to say whether syndecan 1 is expressed on the hASC bound to your DHUA, as without this it is not possible to conclude that the syndecan effect could be responsible for any possible anti-thrombotic effect seen. 

Minor comments

1.There is a reference not numbered in the main text (p3 "xx Kai Hsai, 2017")

2. Details needed on the fluorescent microscope used ("Zeiss xx") and the objectives used and the excitation and emission wavelengths utilised.

3.Abcam have 23 human syndecan1 antibodies - which one was used?   

4. Have you attempted a primary-free control stain where you only used the secondary antibody to see if this can rule out a non-specific binding of the fluorescent secondary to your syndecan-stained samples? 

Author Response

 We are glad to receive valuable suggestions and recommendations from expertize reviewers. We adopted all of them in the revised manuscript, highlighted in words and listed our responses below:

Reviewer 1

Comments and Suggestions for Authors

In this paper the authors attempt to examine the impact of sphingosine-1-phosphate on human adipose stem cell recellularisation of decellularised human umbilical arteries.   The data doesn't appear to convincingly  show that hASC can effectively recellularise the DHUA, that the cell-free DHUA is triggering thrombus formation (and thus that hASC block this effect), hASC differentiation to endothelial cells, or that hASC have changes in syndecan expression in response to S1P, or that this is responsible for the asserted anti-thrombotic effect of hASC.  This may be due to the difficulties in my reading of the paper due to the poor English and incomplete figure legends. However, I don't think the conclusions are supported by the date, and would therefore not recommend publication of this articles in the International Journal of molecular sciences at this time.

Major comments. 

The quality of English is very poor throughout which makes assessing this paper effectively very difficult.  There are spelling and grammatical errors in most lines.  This work needs significant rewriting before it is ready for publication  (e.g. in vitro is spelt incorrectly in the title, magnificence is used instead of magnification, constructuin instead of construction).  Figure legends are not always detailed enough to understand what is occuring in the experiments shown.

Ans: 1) Thank you for recommendation. The manuscript has been edited by native English speaker. The misspellings and grammatical errors were corrected.

      2) Figure legends were revised to fulfill what the experiments shown.

Figure 3 - The demonstration of hASC attachment to segments of the decellularised artery is very difficult to interpret.  The work appears to suggest that only a small percentage of the construct has adherent cells attached to it.  It would seem likely that the coverage of the DHUA is pretty minimal at day 2, and there is no similar data shown for day 14.  Without significant coverage of the DHUA it is unlikely that this construct would ever be viable as an implantable model.  Do the authors have other pictures to demonstrate the % of surface coverage achieved? Can they compare to HUVECs to see if these are better cells for tissue engineering as hypothesised in the introduction?  

Ans: Thank you for valuable recommendation. The number of seeded cell was relatively low density initially in order to assess the proliferation. Hence, it was expected that the attached cells didn’t cover the lumen completely. But the number of adhesive HUVEC on DHUA increased after two days of S1P treatment (Left panel, second row). Fluorescence intensity also showed a significant increase in proliferation (cell count) of HUVECs on DHUAs (Right panel, upper, p<0.01). From the figures, ASCs could also adhere on the DHUAs (Left panel, third row) but the number of cell adhesion indeed didn’t increase significantly after two days of S1P treatment (left panel, fourth row). Fluorescence intensity did not reveal a significant increase of proliferation (cell count) of ASCs on DHUA (Right panel, lower, p>0.05). These results demonstrated that S1P can enhance HUVEC attachment and proliferation on the DHUV scaffolds but did not totally have the same effect on hASC. But both HUVECs and hASCs reached about 80% coverage to the DHUVs at 14 adys and there was no significantly difference between the groups in the present S1P or not (data not shown). Revised context at line 269-273.

Figure 4 - the clotting assay is indirect.  It relies on haemoglobin washed from the sample being solely due to thrombus formation.  However the differences may also be due to less adherence of erythrocytes directly to the surface of the DHUA.  This makes it difficult to assess an anti-thrombotic effect using this experiment.  A better, simpler demonstration would be to show the degree of thrombus formation seen either using fluorescent labels of platelets, fibrin and erythocytes, or by histological section on the construct, rather than using an indirect demonstration.

Ans: Thank you for valuable suggestion.

1) When we transferred the blood to the distilled water, the lumen of the vessels was also washed by water to lyse the erythrocytes adhered on the wall. The context was revised at line 191-192.

2) Besides kinetic clotting time assay, we also performed the platelet adhesion test which was able to observe the platelet attached on the surface (by SEM) similar with the method you suggested. We used SEM to observe the platelets instead of fluorescent label platelets because SEM could also observe the graft inner surface. However, we thought that these two methods both can quantity the thrombus in the lumen in vitro.

3) Kinetic clotting time and platelet adhesion assays were both used to evaluate the coagulation status of TEVGs in vitro.

Can you observe hASCs in your SEM pictures in figure 5?  

Ans: Thanks for suggestions. We placed indicated arrows on the images to indicate hASCs and platelets. Red arrows indicated hASCs while yellow arrows indicated platelets. Please see the revised figure 5. The revised sentences were added to the legend of Figure 5 in Line 306-310.

Figure 5 -  Although there appears to be a single erythrocyte adherent to the surface in the SEM, there doesn't seem to be an obvious thrombus shown on the cell-free DHUA. How thrombogenic is the DHUA normally? Can you see large scale clotting with a light or fluorescent microscope?  If it doesn't trigger observable blood clotting,it would question how significant the anti-thrombotic effect of the hASC is?  Given the minimal surface coverage of hASC seen in Figure 3, I am unconvinced this is a significant biological effect of S1P in this biological system.

Ans: Thanks for the question and this is one of the limitation in our study. We would like to modify the method according to your suggestion for future study. In this assay, as followed our previous publication (Reference [25]), we did not inject whole blood into the vessel tube. Instead, only platelet rich plasma (PRP) was injected in order to more specifically count the number of platelet attached on the grafts. We thought that PRP may not pure enough to get rid of all the red blood cells, but the residue erythrocytes were very rare. The reason of no thrombus or large scale clotting was observed may be due to that thrombus formation needs the participation of both aggregated platelets and red blood cells.

Although large-scale blood clotting could not be observed in vitro, kinetic clotting time and platelet adhesion assays could be as supplements to demonstrate the effect of ASCs on the coagulation status of TEVGs.

Regarding thrombogenic DHUA, it was shown in our previous publication in Reference [25] for in vitro images and Reference [23]) for in vivo test. Both in vitro and in vivo study demonstrated the thrombogenic property of DHUAs. But this thrombogenecity may result from extracellular matrix exposure after decellularization treatment. When we implanted the decellularized blood vessels into rat aorta, most of rats died due to thrombosis.

In order to show a significant biological effect of S1P, we started from relative low density of cell to tell the difference between cells with and without S1P treatment. Our concern is that EC has the property to stop growth when becoming confluent monolayer in culture. But for implantation test in the future, we would try high density recellularized-grafts to optimize its coverage. Thank you for your suggestion.

The revised sentences were added to the legend of Figure 5 in Line 307-311.

Figure 6C - appears to show CD90 and CD105 are higher in S1P-treated cells, even though the text says they are "turned off".  It is hard to judge this properly though as there is no text in the figure legend to explain what is being shown.  How endothelial-like do the authors consider these s1P-treated hASC cells?  Figure 6B doesn't show cells bearing the cobblestone morphology I would expect of endothelial cells, and appears to be solely on the basis of an increase in CD31 expression. 

Ans:  Thank you for reminding and we agreed suggestion. There were still positive CD90 and CD105 in S1P-treated cells and we deleted the term “turn off” in the context. The term “cobblestone morphology “was also deleted. The expectation of endothelial-like differentiation is solely based on increasing expression of CD31 after S1P treatment at this moment. The results of flow cytometry showed that ASCs turned on CD31 expression and turned off CD29, CD73 expression after five days of culture in S1P-added medium (Figure 6A). There was less expression of CD90 and CD105 in S1P treated hASCs.

For clarification, we emphasized that the study aimed to explore the antithrombotic function of S1P on ASCs with SDC-1 expression instead of induce ASC fully toward mature ECs. Thus, we did not investigate other properties except CD31. But your valuable suggestion will guide our future direction of study.

The context was revised at line 317 to 335 and figure legend was corrected accordingly.

Figure 7.  HUVECs have a relatively low level of basal syndecan expression across all the cells overlapping with the DAPI stain (7A), which is increased by S1P (7b).  In contrast, hASC appear to have limited basal expression of syndecan1 and this appears in discrete spots not overlapping with DAPI staining (7C). Although enhanced in 7D, it still doesn't correlate with all cells stained.  It would appear that this might be either non-specific binding of the secondary or there are heterogenous cell populations with differential expression of syndecan-1.  Given this, it needs exploring is difficult to say whether syndecan 1 is expressed on the hASC bound to your DHUA, as without this it is not possible to conclude that the syndecan effect could be responsible for any possible anti-thrombotic effect seen. 

Ans: Thank you for valuable suggestion. ASCs are a group of heterogeneous cells. It is possible that only part of the cells exhibited SDC-1 expression after S1P treatment.

Minor comments

There is a reference not numbered in the main text (p3 "xx Kai Hsai, 2017")

Ans: The reference was numbered as Reference No. 25. It was revised at line 107.

Details needed on the fluorescent microscope used ("Zeiss xx") and the objectives used and the excitation and emission wavelengths utilised.

Ans: Fluorescent images were obtained with a Zeiss Optimises Axio Imager A1 fluorescence microscope. The context was revised at line 223 to 224

3.Abcam have 23 human syndecan1 antibodies - which one was used?   

Ans: Anti-Syndecan-1 antibody [1A3H4] (ab181789) was used.

Have you attempted a primary-free control stain where you only used the secondary antibody to see if this can rule out a non-specific binding of the fluorescent secondary to your syndecan-stained samples? 

Ans: Yes, for each staining of an antibody, we used PBS instead of 1st antibody and then secondary antibody sample as control to eliminate non-specific signal while setting exposure time on fluorescence microscope.

Reviewer 2 Report

The manuscript is well designed and provide very interesting model for testing anti-trombotic activity. Methodology that is used is modern and adequate for the aim that was set up and results are well presented in a clear manner and it is easy to follow. However, English language must be improved since it is very hard to read and the text is full with typographical errors. This must be corrected (for example, the word "in virto" in the title must be replaced with "in vitro",  "kenetic" must be replaced with "kinetic" at a lot of places and there are other words as well). 

Images in Figures 1, 2, 3 and 7 should be larger since cells and differences are not clearly visible when images are too small. Also, in Figure 7 legend is in the lines 320-325 and should be formatted and placed in the continuation of the line 319.

Overall, I think that this manuscript deserve to be published since provides important and interesting results on the mechanism of sphingosine-1-phosphate activity using very interesting model, but after detailed and comprehensive revision and correction of the language and spelling. 

Author Response

 We are glad to receive valuable suggestions and recommendations from expertize reviewers. We adopted all of them in the revised manuscript, highlighted in words and listed our responses below:

Reviewer 2

Comments and Suggestions for Authors

The manuscript is well designed and provide very interesting model for testing anti-trombotic activity. Methodology that is used is modern and adequate for the aim that was set up and results are well presented in a clear manner and it is easy to follow. However, English language must be improved since it is very hard to read and the text is full with typographical errors. This must be corrected (for example, the word "in virto" in the title must be replaced with "in vitro",  "kenetic" must be replaced with "kinetic" at a lot of places and there are other words as well). 

Ans: Thanks for recommendation. The correction of type errors was done. The full text has been re-edited by native English speaker. The misspellings and grammatical errors were revised in the context.

Images in Figures 1, 2, 3 and 7 should be larger since cells and differences are not clearly visible when images are too small. Also, in Figure 7 legend is in the lines 320-325 and should be formatted and placed in the continuation of the line 319.

Ans: We agreed with your suggestion. The pictures were all 300 dpi according to journal guideline. We enlarged the images in the figures according to your suggestion.

Overall, I think that this manuscript deserve to be published since provides important and interesting results on the mechanism of sphingosine-1-phosphate activity using very interesting model, but after detailed and comprehensive revision and correction of the language and spelling. 

Ans: Thank you for the comment. We revised the manuscript with reviewers’ valuable suggestion to improve readability of our study.

Reviewer 3 Report

This research manuscript describes the anti-thrombotic effect of sphingosine-1-phosphate (S1P) on human adipose-derived stem cells (ASCs) cultured in decellularized human umbilical artery (DHUA) for vascular tissue engineering. The manuscript is interesting and well-written. However, there are many things that the authors need to address.

Page 4 lines 169-171: “The grafts was immersed in the culture medium for 6 h at 37 °C while rotating 360° around its longitudinal axis. Another cell suspension was then added in the same manner, and the graft was rotated 90° around its longitudinal axis.” How did the authors rotate the graft and what was the rotation rate? Why the rotation degree for seeding of ASCs in DHUA was different from that of HUVECs? Please provide the data for tri-lineage (adipogenic, osteogenic and chondrogenic) differentiation of human ASCs. Figure 3: The authors should label the lumen of DHUA in the bright field images. The cells only adhered to certain areas of the luminal surface and the cell density is low. Please explain. The authors should include the bright field and fluorescent images for cross section of lumen with cells pre-stained with CM-DIL. Figure 5: The authors should label the platelets, HUVECs and ASCs in the SEM images. The figure caption is incomplete. Please revise. Caption for figures 6 is incomplete. Please revise. Page 12 Lines 394 – 396: “It is known that SDC-1 could aggregate with many coagulation factors, thus activate platelets and leukocytes to cause thrombi occluding the vessel lumen.” This statement is misleading. Many studies have reported that SDC-1 shedding is important in clearance of pro-inflammatory chemokines, resulting in reduced recruitment and adhesion of platelets and leukocytes to endothelial cells. Page 12 Lines 377 – 378: “Moreover, Nincheri et al., revealed S1P could dose-dependently stimulate ASCs toward smooth muscle cells differentiation.” Do S1P-treated ASCs in this study also differentiate into smooth muscle cells? There are many typo errors. For example, virto (title), prolifereation (page 2 line 76), plateds (page 3 line 130), importantnt (page 11 line 344) and gaol (page 12 line 360) etc.

Author Response

 We are glad to receive valuable suggestions and recommendations from expertize reviewers. We adopted all of them in the revised manuscript, highlighted in words and listed our responses below:

Reviewer 3

Comments and Suggestions for Authors

This research manuscript describes the anti-thrombotic effect of sphingosine-1-phosphate (S1P) on human adipose-derived stem cells (ASCs) cultured in decellularized human umbilical artery (DHUA) for vascular tissue engineering. The manuscript is interesting and well-written. However, there are many things that the authors need to address.

Page 4 lines 169-171: “The grafts was immersed in the culture medium for 6 h at 37 °C while rotating 360° around its longitudinal axis. Another cell suspension was then added in the same manner, and the graft was rotated 90° around its longitudinal axis.” How did the authors rotate the graft and what was the rotation rate? Why the rotation degree for seeding of ASCs in DHUA was different from that of HUVECs?

Ans: We injected the cell suspension to grafts and immersed in the culture medium incubate for 6 h in culture dish, repeated the procedure 4 times by rotate 90° each, total 360°.

Please provide the data for tri-lineage (adipogenic, osteogenic and chondrogenic) differentiation of human ASCs.

Ans: The differentiation potential of ASCs to adipogenic, osteogenic and chondrogenic linage was examined using a differentiation–induction protocol and differentiation assay described previously [29, 30] and the result was shown in supplement data.

Figure 3: The authors should label the lumen of DHUA in the bright field images. The cells only adhered to certain areas of the luminal surface and the cell density is low. Please explain. The authors should include the bright field and fluorescent images for cross section of lumen with cells pre-stained with CM-DIL.

Ans: We agreed with your suggestion. The lumen was labeled in the image of Figure 3.

In order to show a significant biological effect of S1P, we started from relative low density of cell to tell the difference between cells with and without S1P treatment. Our concern is that EC has the property to stop growth when becoming confluent monolayer in culture. But for implantation test in the future, we would try high density recellularized-grafts to optimize its coverage

Figure 5: The authors should label the platelets, HUVECs and ASCs in the SEM images. The figure caption is incomplete. Please revise.

Ans: Thanks for suggestion. We labeled the platelets on Figure 5 and completed the figure legend. The manuscript was revised at line 307 to 311.

Caption for figures 6 is incomplete. Please revise.

Ans: The figure legend of figures 6 was completed as below:  

A: After culture in S1P-added culture medium, ASCs showed positive CD31 and negative CD29, CD73. The expressin of CD90 and CD105 on hASC became less. It indicated that ASCs differentiated towards EC. B: The morphology of ASCs under S1P-added culture medium (magnification 100×, scale bar = 10 μm).

The manuscript was revised at line 332 to 335.

Page 12 Lines 394 – 396: “It is known that SDC-1 could aggregate with many coagulation factors, thus activate platelets and leukocytes to cause thrombi occluding the vessel lumen.” This statement is misleading. Many studies have reported that SDC-1 shedding is important in clearance of pro-inflammatory chemokines, resulting in reduced recruitment and adhesion of platelets and leukocytes to endothelial cells.

Ans: Thanks for the correction. To avoid confusion, we deleted this phrase and citation accordingly.

Page 12 Lines 377 – 378: “Moreover, Nincheri et al., revealed S1P could dose-dependently stimulate ASCs toward smooth muscle cells differentiation.” Do S1P-treated ASCs in this study also differentiate into smooth muscle cells?

Ans: In Nincheri’s study, the minimum concentration of S1P used to induce differentiation of ASC towards smooth muscle cell was 0.1 uM (after six days). Under 1 uM S1P treatment for 6 days, profound cytoskeletal reorganization was noted in ASCs. However, the culture medium used for smooth muscle differentiation was DMEM without serum containing 1mg/ml BSA and not endothelial cell growth medium used in our study. Thus, we believed S1P-treated ASCs in our study wouldn’t differentiate into smooth muscle cells but endothelial cells. The flow-cytometry also showed CD31 shifting to indicate the possibility of endothelial-like cell differentiation.

There are many typo errors. For example, virto (title), prolifereation (page 2 line 76), plateds (page 3 line 130), importantnt (page 11 line 344) and gaol (page 12 line 360) etc.

   Ans: The correction of type errors was done. The full text has been edited by native English speaker. The misspellings and grammatical errors were correct. Such as in vitro (title) at line 5, proliferation (previous page 2 line 76) was deleted, plate (previous page 3 line 130) at line 133, page 3, important (previous page 11 line 344) at line 371, page 12, goal (previous page 12 line 360) at line 388, page 12.

Round 2

Reviewer 3 Report

The authors have addressed my comments. I have no further comments.

Author Response

No questions to response